# Inequalities in Psychiatric Morbidity in Hong Kong and Strategies for Mitigation

**DOI:** 10.3390/ijerph19127095

**Published:** 2022-06-09

**Authors:** Siu-Ming Chan, Linda Chiu-Wa Lam, Wing-Yan Law, Se-Fong Hung, Wai-Chi Chan, Eric Yu-Hai Chen, Gary Ka-Ki Chung, Yat-Hang Chan, Roger Yat-Nork Chung, Hung Wong, Eng-Kiong Yeoh, Jean Woo

**Affiliations:** 1CUHK Institute of Health Equity, The Chinese University of Hong Kong, Hong Kong SAR, China; siuming.chan@cityu.edu.hk (S.-M.C.); gchung@cuhk.edu.hk (G.K.-K.C.); lucachan@cuhk.edu.hk (Y.-H.C.); rychung@cuhk.edu.hk (R.Y.-N.C.); hwong@cuhk.edu.hk (H.W.); yeoh_ek@cuhk.edu.hk (E.-K.Y.); 2Department of Social and Behavioural Sciences, City University of Hong Kong, Hong Kong SAR, China; 3Department of Psychiatry, The Chinese University of Hong Kong, Hong Kong SAR, China; cwlam@cuhk.edu.hk (L.C.-W.L.); yanlaw@cuhk.edu.hk (W.-Y.L.); sefonghung@gmail.com (S.-F.H.); 4Department of Psychiatry, The University of Hong Kong, Hong Kong SAR, China; waicchan@hku.hk (W.-C.C.); eyhchen@hku.hk (E.Y.-H.C.); 5CUHK Institute of Ageing, The Chinese University of Hong Kong, Hong Kong SAR, China; 6School of Public Health and Primary Care, The Chinese University of Hong Kong, Hong Kong SAR, China; 7Department of Social Work, The Chinese University of Hong Kong, Hong Kong SAR, China

**Keywords:** mental morbidity, social gradient, inequality, mental health policy, Hong Kong

## Abstract

This study explores the social gradient of psychiatric morbidity. The Hong Kong Mental Morbidity Survey (HKMMS), consisting of 5719 Chinese adults aged 16 to 75 years, was used. The Chinese version of the Revised Clinical Interview Schedule (CIS-R) was employed for psychiatric assessment of common mental disorders (CMD). People with a less advantaged socioeconomic position (lower education, lower household income, unemployment, small living area and public rental housing) had a higher prevalence of depression and anxiety disorder. People with lower incomes had worse physical health (OR 2.01, 95% CI 1.05–3.82) and greater odds of having CMD in the presence of a family history of psychiatric illnesses (OR 1.67, 95% CI 1.18–2.36). Unemployment also had a greater impact for those in lower-income groups (OR 2.67; 95% CI 1.85–3.85), whereas no significant association was observed in high-income groups (OR 0.56; 95% CI 0.14–2.17). Mitigating strategies in terms of services and social support should target socially disadvantaged groups with a high risk of psychiatric morbidity. Such strategies include collaboration among government, civil society and business sectors in harnessing community resources.

## 1. Introduction

### 1.1. Background

Public health and social policy studies report growing concern regarding individual mental health. The increasing burden and consequences of mental health problems have been widely documented in the existing literature [1,2,3,4,5]. For example, the risk of all-cause mortality was doubled among those with mental health problems [6]. This growing burden results in significant social and economic loss, which highlights the urgency for policymakers to devise effective policies to reduce the adverse societal effect of mental health problems. On the other hand, there has been increasing attention on the social determinants of health, including but not limited to age, gender, education, employment status, income and other more upstream societal factors, which could, in turn, shape our health [7,8,9]. Differential exposure to these social determinants of health would result in social gradient or inequality across the social ladder, with extensive evidence supporting the fact that individuals with lower socioeconomic position tend to not only have worse physical health but also poorer mental health conditions [10,11]. Plausible mechanisms behind the mental health impact of socioeconomic disadvantage include stress stemming from financial hardship, upward social comparison with the wealthier counterparts, as well as a lower sense of security, control and social connectedness [12]. Nonetheless, inequalities in mental health and related mitigating measures have received comparatively less attention from policymakers across countries. A recent report in the UK showed that there have been substantial and persistent mental health inequalities among children over the past 20 years, despite government commitments to improve mental health [13], whereas another study also revealed that mental health inequalities exist even in relatively egalitarian countries in Europe [14].

### 1.2. The Context of Hong Kong

The Hong Kong Special Administrative Region of China (HKSAR), a highly developed and densely populated society, is facing growing concern regarding mental disorders, as well as income inequality and poverty issues [15,16,17]. Recent studies identified some key social determinants of mental health, such as housing, income and deprivation using generic quality-of-life measures [18,19,20,21,22]. However, few studies examined mental health inequalities in terms of psychiatric diagnosis.

Currently, the Hong Kong Government has implemented a number of policies and social services for mental health. Approximately 8% of the total health budget is allocated to mental health. The Hospital Authority (HA), as the major service provider by providing multidisciplinary professional services in Hong Kong, has started employing a case management model in recent years to provide personalized and patient-centred care programmes in the community. Moreover, Public–Private Partnership programmes have also been initiated to alleviate the long waiting time for public psychiatric specialist outpatient services [23]. To enhance the well-being of employees, the Department of Health and Labour launched the ″Joyful@Healthy Workplace” programme in 2016 to create a healthy working environment for employees and employers. Companies were encouraged to sign a workplace charter in promoting mental well-being and healthy workforce in workplaces. In addition to the effort by the government, non-government organizations (NGOs) and charitable foundations play a significant role in providing social services and support to people facing mental health problems. The Integrated Community Centers for Mental Wellness (ICCMWs), funded by the government and operated by different NGOs, have been operating since 2010 to provide convenient community services, ranging from prevention for discharged mental patients to persons with suspected mental health problems. On the other hand, the Hong Kong Jockey Club Charities Trust, one of the largest charities in Hong Kong, donated more than HKD 3.7 billion in the community in 2019/20, with a number of community projects related to mental health promotion, including public education, suicide prevention programmes, a holistic support project for elderly mental wellness and an adolescent mental health centre. Furthermore, the business sectors also actively contributed to care work in the community, such as providing volunteer social services, public education and building up social capital. Nevertheless, the mental health policy is fragmented, with little coordination across government, civil society and business sectors.

Despite the increase in the mental health budget and supporting services, psychiatric morbidity remains a serious problem. In 2020, there were 44,541 new cases at psychiatry specialist outpatient clinics. The median waiting time of stable cases ranged from 15 to 46 weeks across districts, while the longest (90th percentile) waiting time was up to 100 weeks in the New Territories East [24]. In 2018–2019, more than 250,000 psychiatric patients were treated in HA; however, the work force in psychiatry was limited. The nurse-to-patient and doctor-to-patient ratios per 1000 inpatient and day-patient discharges and deaths in psychiatry were 138.6 and 19.4, respectively [23].

### 1.3. Aims of Study

Although there was an increase in resources for health and social services, aiming at enhancing the mental health of individuals in Hong Kong in recent years, mental health policies have not focused on mental health inequalities across the social ladder, except some previous efforts on identifying vulnerable groups for targeted services. Hence, this study aims to examine whether there is a social gradient in psychiatric morbidity and potential mitigating factors, using data from the Hong Kong Mental Morbidity Survey (HKMMS), the largest territory-wide epidemiological study of psychiatric morbidity, conducted between 2010 and 2013 in Hong Kong. We hypothesized that there is a social gradient of psychiatric morbidity and that socioeconomic factors have a significant impact on the mental health of individuals.

## 2. Method

### 2.1. Data and Sample

The HKMMS is the first and only territory-wide psychiatric epidemiological study with representative population sampling in Hong Kong. A stratified and multi-stage sampling design was employed, which was modelled on the British Adult Psychiatric Morbidity Survey (APMS). Face-to-face interviews were conducted by professionally trained interviewers between November 2010 and May 2013. In total, 5719 Chinese adults aged 16 to 75 years in Hong Kong were successfully interviewed. The HKMMS study aims to assess the overall prevalence rates and associate factors of different CMD including depression, generalized anxiety disorder and other anxiety disorders. For more details on the methodology, please refer to previous papers based on the HKMMS [25,26].

### 2.2. Measurement

#### 2.2.1. Common Mental Disorders (CMD)

The Chinese version of the Revised Clinical Interview Schedule (CIS-R), which is a structured psychiatric assessment tool [27], was used to assess non-psychotic symptomatic morbidity of participants in the week prior to interview. The diagnosis of CMD was based on the International Classification of Diseases (ICD-10). The validated cutoff score of 12 [27] was used to identify those cases with CMD in the sample [25]. Specially, the prevalence of depression (DEP), generalized anxiety disorder (GAD), mixed anxiety and depressive disorder (MADD) and other anxiety disorders (OAD) were examined in this study, according to the ICD-10.

#### 2.2.2. Socioeconomic Variables

The social gradients of various mental disorders were examined by their socioeconomic position (SEP), in terms of education level, employment status, income, perceived financial difficulties, housing type and living floor area. Education level was divided into four groups, ranging from ″no schooling/primary″ to ″post-secondary″; employment status was divided into ″working″, ″economically inactive” and ″unemployed/not working”; household income was grouped into ″below HKD 15,000 (low)”, ″HKD 15,000–39,999 (middle)” and ″above HKD 40,000 (high)”. There were three main types of housing, including ″public rental housing”, ″subsidized home ownership housing” and ″private permanent housing”. The living floor area was divided into tertiles for analysis. In addition, the participants were asked whether they perceived having financial difficulties (or subjective poverty) with answer ″yes” or ″no”. These variables will also be used in logistic regression analysis.

#### 2.2.3. Confounding Variables for Logistic Regression Analysis

The risk factors of psychiatric morbidity were examined by logistic regression, with adjustment for various confounding factors including demographic variables, physical health, lifestyle factors, family history of psychiatric illnesses and stressful life events. The demographic factors included gender, age, education level, marital status, employment status and household size. The physical health was assessed by the Cumulative Illness Rating Scale (CIRS). Participants who rated 3 (severe impairment) or above in one of the 13 physical illness domains were identified as having severe impairment [28]. About lifestyle factors, smoking status was categorized as never smoker, smoker and ex-smoker. Alcohol drinking was categorized as hazardous/harmful drinking or no hazardous/harmful drinking, assessed by Alcohol Use Disorders Identification Test [29] and Community version of the Severity of Alcohol Dependence Questionnaire [30]. Substance dependence was assessed by a questionnaire on substance misuse and dependence. Moreover, participants were asked whether they had family history of psychiatric illnesses and stressful life events. A 17-item checklist of stressful life events, which has been shown to be associated with psychological distress, was used for the measurement [31].

#### 2.2.4. Perceived Social Support

Social support was one of the main independent variables in this study of mental morbidity. The 12-item Multidimensional Scale of Perceived Social Support (MSPSS) was used to assess the perception of social support from the family, friends and significant others. Participants were asked to rate on a 7-point scale, where the sum total of the item scores ranges from 12 to 84 [32]. For logistic regression, the total score of MSPSS was divided into tertiles.

### 2.3. Statistical Analytical Strategy

The statistical analysis was divided into two parts. The first part studied whether there is a social gradient of CMD in Hong Kong. The prevalence of CMD, including DEP, GAD, MADD and OAD, was examined by various socioeconomic factors. The percentages of prevalence were compared across categories of the socioeconomic variables using ANOVA to examine the significance of differences among groups. The second part investigated the risk and protective factors of CMD by income groups. Univariate analyses on the crude associations of risk factors and potential confounders with CMD were first conducted. Multivariable binary logistic regressions were then performed to examine the associations of socioeconomic factors and perceived social support with CMD, with adjustment for confounding variables. Weighting was applied for age and gender adjusted to data from 2011 Hong Kong Population Census. All statistical tests were two-tailed with a significance level of *p* < 0.05. Data were analysed using Statistical Package for the Social Sciences (SPSS) v26.

## 3. Result

### 3.1. Prevalence of CMDs by Socioeconomic Factors

The one-week prevalence of CMD, DEP, GAD, MADD and OAD are reported in Table 1. Those having a low education level (i.e., no school or primary level) had a higher prevalence of DEP (6.7%) and GAD (7.1%) than those with higher education levels (*p* < 0.001). In terms of employment status, the ″unemployed/not working” cases had a higher prevalence of all CMDs compared with the working population and economically inactive cases. The prevalence of DEP, GAD, MADD and OAD of the unemployed group was 15.1%, 12.2%, 9.8% and 5.4%, respectively, much higher than the prevalence of those CMDs in the working group (*p* < 0.001). For household income, the prevalence of DEP (6.3%), GAD (7.3%) and OAD (3.1%) of the low-income group was much higher than that of the high-income groups (*p* < 0.001). The participants who perceived themselves having financial difficulties also had a significantly higher (*p* < 0.001) prevalence of CMD (31.5%) than those with no financial difficulties (9.8%). Considering housing type, residents in public rental housing had a higher prevalence of CMD than those living in subsidized housing and private permanent housing (*p* < 0.001). Overall, participants who are less advantaged in terms of SEP, including lower education, unemployment, lower household income, perceived financial difficulties, living in public rental housing and small living area, had a higher prevalence of DEP, GAD and OAD than those with better SEP.

### 3.2. Risk and Protective Factors of CMD by Income Group

The crude and adjusted odds ratios (OR), with their corresponding 95% confidence intervals (CI) and p values, of the associations with CMD are displayed in Table 2. In univariate models, being female, divorced/separated/widowed, hazardous drinking and substance-dependent, as well as those having stressful life events, perceived financial difficulties and low social support, were significantly associated with CMD in all income groups. In multivariable models, with an adjustment in demographic variables, being female, substance-dependent and those having perceived financial difficulties, stressful life events and low perceived social support showed significant impacts on CMD in all income groups.

For income group comparison, the effect of age was significant in the low-income group; those aged 40 to 59 (OR 0.64; 95% CI 0.44–0.93) and aged 60 or above (OR 0.47; 95% CI 0.29–0.76) had lower risk of CMD, compared with those aged 16 to 39 in the low-income group. The adjusted effect of ″unemployed/not working″ (OR 2.67; 95% CI 1.85–3.85) was higher in the low-income group than other income groups, and not significant in the high-income group (OR 0.56; 95% CI 0.14–2.17), implying the importance of job security in enhancing the mental well-being of low-income households. Physical health (OR 2.01, 95% CI 1.05–3.82) and family history of psychiatric illnesses (OR 1.67, 95% CI 1.18–2.36) were significantly associated with CMD, only in the low-income group and not in the other two income groups. Moreover, being divorced, separated and widowed (OR 3.17, 95% CI 1.18–8.53) and being a smoker (OR 2.02, 95% CI 1.05–3.91) or ex-smoker (OR 3.21, 95% CI 1.45–7.13) were significantly associated with CMD in the high-income group only.

Among all income groups, being female, having perceived financial difficulties and low perceived social support were significantly associated with CMD. The adjusted odds ratio of being female was higher in the low-income group (OR 2.98, 95% CI 2.14–4.14) than that in the middle-income (OR 2.75, 95% CI 1.93–3.92) and high-income groups (OR 2.38, 95% CI 1.55–3.66). On the other hand, the effect of perceived financial difficulties was higher in the high-income group (OR 4.63, 95% CI 2.57–8.35) than that in the low-income group (OR 2.78, 95% CI 2.10–3.68). The effect of high social support was slightly smaller in the low-income group (OR 0.22, 95% CI 0.14–0.34) compared with that in the high-income group (OR 0.30, 95% CI 0.18–0.49).

## 4. Discussion

This study is the first territory-wide epidemiological examination of mental health inequality in Hong Kong. The social gradient of mental health was comparable to other developed cities across the world. The descriptive result clearly showed a social gradient of mental health, suggesting that individuals with poorer SEP were at higher risk of psychiatric morbidity. The results echo studies in Western countries that poor social and economic circumstances negatively affect individual mental health. As observed in previous international studies, there is a significant difference in mental health outcomes across SEP, including education, income and employment [33,34], even in relatively egalitarian countries in Europe [14]. For instance, a recent study in Sweden suggested that poor mental health conditions and psychiatric diagnoses have been increasingly concentrated among the least-educated and lowest-income groups [35], whereas another study in the Netherlands supported the association of job loss with mood disorders [36]. Nonetheless, most of these studies did not examine or report the potential moderating effect by income levels. As shown in our findings, the mental health effect of unemployment is particularly strong among lower-income groups than their wealthier counterparts, which may reflect the inadequacy of the existing mental health support policies for the low-income working group in Hong Kong. In addition, our logistic regression analyses also identified other common risk factors of CMD and the specific risk factors corresponding to different income groups. For example, females had a higher risk of psychiatric morbidity than males among all income groups, while older adults had a lower risk of psychiatric morbidity than younger adults in the low-income group. This analysis helps us to evaluate whether the existing mental health policies and services appropriately tackle the mental health inequality problem or target the at-risk population in Hong Kong.

### 4.1. Policy Implication Based on the Social Gradients and Risky Factors of Mental Health

The social policy response should target the social groups with a higher prevalence of psychiatric morbidity, including those with low education level, unemployed, low household income, living in public housing or in small living area. However, the existing government policies overlooked the social gradient of mental health in Hong Kong. For example, although the unemployed had a much higher prevalence of CMD than the working group and the economically inactive group, as reflected by our results, the existing service for the unemployed mainly focused on job seeking and retraining rather than on mental health support. For division of work, the Labor Department is responsible for providing employment services, whereas the SWD or the ICCMW are in charge of mental health supporting services. One suggestion is to employ mental health support elements in the current employment supporting service, such as providing mental health promotion programmes and activities along with the employment service of the Labor Department. The staff in the Labor Department should be well trained to identify the mentally at-risk cases and to refer them to corresponding social services.

On the other hand, being female was a common risk factor of CMD among all income groups in this study; nonetheless, the mental burden of women was commonly overlooked. As the work participation rate of females in Hong Kong increased rapidly in recent years [37], meaning working mothers are facing a growing risk of psychiatric morbidity. Especially in traditional Chinese culture, women also take up a major role in caregiving and housework [38], which further exerts psychological distress on them [39]. Although the Hong Kong Government started adopting recommendations from the Women′s Commission in 2015 and applied gender mainstreaming in formulating government policy, gender perspective was not a key theme in mental health policy in Hong Kong. In the recent mental health review report, there was also little concern on gender issues [40]. In contrast, philanthropic foundations supplemented these service gaps. For example, the Hong Kong Jockey Club Charities Trust (HKJC) initiated the Jockey Club Mental Wellness Project for Women, which provided targeted service to women with consultation, educational talks, group activities and integrated services. The targeted project acknowledged the pressure from work and family faced by women. This example highlighted the important role of the third sector in mitigating mental health inequality in Hong Kong. These projects initiated by the HKJC were proven as effective in enhancing the happiness and mental health of the project participants [41], which could serve as a crucial reference for government policy design in the future. In terms of age, our results showed that those aged 16 to 39 had a higher risk of CMD compared with the older groups, and this finding was significant in the low-income group. One possible explanation is that there were relatively multitudinous social services and community centres targeting low-income older adults so that most of them can receive stable financial support from the government. Nevertheless, a previous study showed that the older adults generally had a higher prevalence of DEP and GAD but lower prevalence of MADD and OAD than younger adults [26]. The mechanism behind the difference in prevalence of various psychiatric morbidity among age groups warrants in-depth studies in the future.

Specifically, low-income families were more vulnerable to psychiatric morbidity. Within the low-income group, risk factors should be highlighted and more targeted services and resources are needed. First, cases with physical illness or morbidities were the potential at-risk group for targeted interventions. The existing medical service tended to tackle the physical and mental health problems separately; however, the health needs of these individuals are often complex, requiring holistic care. For example, the District Health Centers (DHC) responsible for primary healthcare could liaise with the ICCMW in the community. More integrated services that promote physical and mental health as a one-stop service are also encouraged. Second, our result found that low-income participants with a family history of psychiatric illness were at a higher risk of psychiatric morbidity. More resources should be allocated to them with a targeted follow-up case management service. In addition, it is worth noting that mental morbidity may lead to a worsening of social and economic position. Persons with mental morbidity are more vulnerable to unemployment. Policy intervention on income enhancement, inclusive employment and mental health promotion would be crucial to prevent a vicious cycle of downward social drift.

Furthermore, our results demonstrated that social support was a crucial protective factor of CMD across all income groups. Promoting mutual help and building up social capital are essential for mitigating mental health inequality. The Community Investment and Inclusion Fund (CIIF) is one of the key government policies in promoting social capital in the community. Although the Hong Kong Government increased the budget up to HKD 500 million in 2019, the social needs and risks in the community are growing rapidly at the same time. The overall budget for the social welfare service, including resources to the ICCMW, integrated family service centres (IFSC), youth and elderly community centres, should be increased progressively to enhance social cohesion and collectiveness in the community. On the other hand, perceived financial difficulty was a common risk factor of psychiatric morbidity, especially in the high-income group. The high living expenses in Hong Kong, including housing expenditure, education and medical expenses, were a plausible explanation. A recent worldwide survey showed that Hong Kong is the world’s most expensive city, with the highest cost of living [42]. This result also echoed a previous study showing that high housing affordability affects the mental health of individuals in Hong Kong [18]. Strategies and policies to tackle the high living cost are crucial for reducing the risk of psychiatric morbidity.

Overall, existing mental health policies in Hong Kong did not include an agenda on tackling mental health inequality. The newly established Advisory Committee on Mental Health (ACMH) should design relevant policies on reducing mental health inequality as well as more social services targeted to social groups at risk, including women, the unemployed and those with little social support in the community. Moreover, the involvement of the business sector could play an important role in mitigating societal mental health inequalities. For example, the Caring Company Scheme, launched by the Hong Kong Council of Social Service (HKCSS) in 2002, promotes the connection between the business and social sectors, with active ageing and community mental health as the two key themes. More than 4000 companies and 480 social service organizations joined the scheme in 2019/20, which provided 220,000 h of volunteering services. Additional effort should be put on collaboration among government, civil society and business sectors. Furthermore, government can take reference of the projects initiated by civil society, the business sector and NGOs with positive evaluation results [43], to plan more long-term mental health services in the future.

Given the time lag of our study, the observed apparent social gradients of mental illnesses could be further exacerbated by social unrest and the ongoing COVID-19 pandemic [44], which is alarming and warrants policy actions. Recent studies found that the COVID-19 pandemic had serious impacts on individual mental health and socioeconomic inequality as mental health still exists during the pandemic, both internationally [45,46] and in Hong Kong [47,48,49,50,51]. Specifically, deprived individuals had worse mental health scores via concerns over livelihood and economic activity (e.g., unemployment or job instability), which were likely to be exacerbated by the stringent COVID-19 containment measures in Hong Kong [47,48]. As the social context and people’s vulnerability to mental illnesses are changing over time, it is suggested that the government should initiate continuous studies on social gradients of the prevalence and incidence of psychiatric morbidity in order to monitor the mental health situation and the impact of mental health policy implementation in Hong Kong. Psychiatric morbidities could be included as one of the health outcomes in regularly collected data by the government in monitoring health inequalities.

### 4.2. Limitations

There are several limitations in the current study. First, the cross-sectional nature of the HKMMS could not establish a causal relationship among the variables. Longitudinal research is, therefore, recommended to study the change in the social gradient of mental morbidity after the implementation of mental health policies and services. Second, some of the survey questions, such as social support and financial difficulties, are self-reported and, thus, may be subject to recall bias. Third, the HKMMS was conducted between 2010 and 2013, which may not be able to reflect the latest local situation. Nevertheless, the HKMMS was the only and territory-wide survey of psychiatric morbidity in Hong Kong. Moreover, the CIS-R measurement adopted in the HKMMS provided a more accurate assessment of psychiatric morbidity prevalence than any other mental health prevalence studies in Hong Kong.

## 5. Conclusions

In conclusion, this is the first study in Hong Kong that examined inequalities in psychiatric morbidity, risk factors and potential mitigating strategies. People who are less advantaged in terms of socioeconomic position (unemployment, lower household income and education) have a higher prevalence of common mental disorders. Moreover, women and individuals with less social support and perceived financial difficulties were at a higher risk of psychiatric morbidity. However, existing mental health policies do not address such inequalities. An increase in government mental health services and resources could target social groups at higher risk, working in a coordinated way with civil society and business sectors, in order to maximize resource utilization towards the mitigation of inequalities in psychiatric morbidity.

## Figures and Tables

**Table 1 ijerph-19-07095-t001:** Descriptive result of social gradient in psychiatric morbidity.

		Prevalence % of Psychiatric Morbidity									
	N	Any CMD		DEP		GAD		MADD		OAD	
		%	*p*-Value	%	*p*-Value	%	*p*-Value	%	*p*-Value	%	*p*-Value
**Education**			<0.001		<0.001		<0.001		0.226		0.008
No schooling/Primary	733	17.9		6.7		7.1		6.8		1.5	
Lower secondary	982	15.0		4.0		4.9		7.0		2.4	
Upper secondary	2631	13.8		2.5		4.0		7.7		1.8	
Post-secondary	1371	10.9		1.6		3.7		5.9		0.7	
**Employment Status**			<0.001		<0.001		<0.001		0.021		<0.001
Working	3510	11.4		1.6		3.5		6.5		1.1	
Economically inactive (Retired/ Housewife/Student)	1723	13.8		2.8		4.1		7.5		1.6	
Unemployed/ Not working	483	32.1		15.1		12.2		9.8		5.4	
**Household Income (in Hong Kong dollar)**			<0.001		<0.001		<0.001		0.012		<0.001
Below $15,000	1907	20.1		6.3		7.3		8.2		3.1	
$15,000–39,999	2241	10.1		1.2		2.9		5.9		0.9	
Above $40,000	1241	11.1		1.6		2.6		7.7		0.6	
**Perceived financial difficulties**			<0.001		<0.001		<0.001		<0.001		<0.001
No	4659	9.8		1.5		2.9		5.7		1.0	
Yes	1057	31.5		10.0		11.5		13.1		4.2	
**Housing Type**			<0.001		<0.001		0.002		0.689		0.001
Public rental housing	2323	16.2		4.3		5.6		7.4		2.4	
Subsidized home ownership housing	921	11.9		1.8		3.6		6.7		0.8	
Private permanent housing	2473	12.3		2.3		3.7		6.8		1.2	
**Living Floor Area**			<0.001		<0.001		<0.001		0.138		<0.001
Lower tertile	1766	17.6		5.4		6.2		7.6		2.8	
Middle tertile	2014	13.0		2.2		3.8		7.5		1.1	
Higher tertile	1877	10.9		1.7		3.5		6.1		1.0	

Note. CMD: common mental disorder; DEP: depressive episode; GAD: generalized anxiety disorder; MADD: mixed anxiety and depressive disorder; OAD: other anxiety disorder; weights were applied for age and gender.

**Table 2 ijerph-19-07095-t002:** Associations between CMD, socio-demographic background and other key factors, by income groups.

	CMD					
	Low Income: <$15,000		Middle Income: $15,000 to $39,999		High Income: >$40,000	
	Crude OR	Adjusted OR ^a^	Crude OR	Adjusted OR ^a^	Crude OR	Adjusted OR ^a^
**Gender**						
Male	1	1	1	1	1	1
Female	2.10(1.65–2.68) ***	2.98(2.14–4.14) ***	2.27(1.68–3.05) ***	2.75(1.93–3.92) ***	1.50(1.05–2.15) *	2.38(1.55–3.66) ***
**Age**						
16 to 39	1	1	1	1	1	1
40 to 59	1.11(0.85–1.44)	0.64(0.44–0.93) *	0.66(0.49–0.89) **	0.49(0.33–0.72) ***	0.89(0.62–1.27)	0.75(0.48–1.17)
≥60	0.72(0.53–0.97) *	0.47(0.29–0.76) **	0.66(0.39–1.13)	0.54(0.27–1.10)	0.56(0.21–1.50)	0.43(0.13–1.39)
**Education**						
No schooling/Primary	1	1	1	1	1	1
Lower secondary	0.89(0.65–1.21)	0.83(0.57–1.22)	0.93(0.51–1.71)	1.00(0.50–2.01)	0.14(0.01–2.33)	0.18(0.01–3.80)
Upper secondary	0.88(0.66–1.16)	1.08(0.74–1.57)	1.06(0.63-1.80)	0.96(0.50-1.84)	1.32(0.33–5.31)	1.11(0.18–6.89)
Post-secondary	0.63(0.39–1.04)	1.13(0.61–2.12)	1.12(0.63–1.99)	0.84(0.39–1.77)	0.96(0.24–3.86)	0.97(0.15–6.21)
**Marital Status**						
Married/Cohabit	1	1	1	1	1	1
Single	0.80(0.60-1.06)	0.59(0.39–0.89) *	1.13(0.84–1.52)	0.76(0.51–1.13)	1.03(0.69–1.54)	1.00(0.60–1.69)
Divorced/ Separated/Widowed	2.12(1.63–2.77) ***	1.22(0.86–1.74)	1.79(1.09–2.91) *	1.26(0.71–2.23)	3.63(1.60–8.22) **	3.17(1.18–8.53) *
**Employment Status**						
Working	1	1	1	1	1	1
Economically inactive (Retired/Housewife/Student)	1.25(0.95–1.63)	1.39(0.99–1.95)	0.95(0.68–1.32)	0.99(0.66–1.48)	0.93(0.56–1.55)	0.73(0.38–1.38)
Unemployed/Not working	3.60(2.66–4.88) ***	2.67(1.85–3.85) ***	2.85(1.74–4.68) ***	2.43(1.38–4.28) **	0.62(0.18–2.15)	0.56(0.14–2.17)
**Household Size**						
1-person	1	1	1	1	1	1
2-person	0.84(0.62–1.13)	0.96(0.66–1.40)	0.62(0.38–1.03)	0.68(0.37–1.24)	0.64(0.30–1.34)	0.87(0.36–2.13)
3-person	0.82(0.60–1.12)	0.90(0.59–1.38)	0.43(0.26–0.69) **	0.46(0.26–0.84) *	0.85(0.42–1.75)	1.47(0.61–3.55)
4-person or more	0.71(0.50–1.00)	0.94(0.57–1.54)	0.55(0.35–0.87) *	0.65(0.36–1.19)	0.54(0.27–1.10)	0.84(0.35–1.99)
**Cumulative Illness Rating severe impairment**						
No	1	1	1	1	1	1
3/4 in any domain	3.04(1.91–4.98) ***	2.01(1.05–3.82) *	2.34(0.62–8.83)	1.91(0.38–9.49)	1.48(0.27–8.20)	1.80(0.26–12.29)
**Smoking**						
Never-smoker	1	1	1	1	1	1
Smoker	1.33(1.00–1.78)	1.20(0.80–1.79)	1.19(0.82–1.74)	0.95(0.58–1.54)	2.38(1.44–3.92) **	2.02(1.05–3.91) *
Ex-smoker	0.79(0.52–1.2)	0.91(0.53–1.55)	1.04(0.62–1.73)	1.31(0.73–2.35)	2.97(1.49–5.93) **	3.21(1.45–7.13) **
**Alcohol**						
No hazardous/ harmful drinking	1	1	1	1	1	1
Hazardous/harmful drinking	1.95(1.25–3.05) **	1.27(0.70–2.30)	1.99(1.27–3.12) **	1.95(1.12–3.39) *	2.22(1.14–4.29) *	1.48(0.65–3.38)
**Substance Dependence**						
No	1	1	1	1	1	1
Yes	3.88(2.19–6.88) ***	2.35(1.18–4.67) *	2.43(1.16–5.09) *	2.41(1.06–5.49) *	4.76(1.94–1.67) **	3.10(1.10–8.72) *
**Family history of** **psychiatric illnesses**						
No	1	1	1	1	1	1
Yes	1.77(1.32–2.37) ***	1.67(1.18–2.36) **	1.72(1.20–2.47) **	1.44(0.97–2.14)	1.51(0.97-2.36)	1.34(0.80-2.22)
**Life event (any)**						
No	1	1	1	1	1	1
Yes	2.81(2.10-3.76) ***	2.15(1.54-3.01) ***	2.74(1.95-3.84) ***	2.58(1.79-3.72) ***	3.21(1.95-5.29) ***	2.86(1.69-4.84) ***
**Perceived Financial** **Difficulties**						
No	1	1	1	1	1	1
Yes	4.23(3.35–5.35) ***	2.78(2.10–3.68) ***	3.06(2.24–4.18) ***	2.47(1.74–3.50) ***	4.8(2.89–7.99) ***	4.63(2.57–8.35) ***
**Housing Type**						
Public rental housing	1	1	1	1	1	1
Subsidized home ownership housing	0.55(0.37–0.82) **	0.75(0.45–1.23)	1.32(0.91–1.91)	1.39(0.88–2.18)	1.29(0.60–2.76)	1.99(0.78–5.08)
Private permanent housing	0.73(0.56–0.95) *	0.81(0.57–1.17)	1.15(0.84–1.57)	1.21(0.82–1.80)	1.48(0.78–2.81)	2.92(1.24–6.86) *
**Living Floor Area**						
Lower tertile	1	1	1	1	1	1
Middle tertile	0.57(0.43–0.75) ***	0.83(0.58–1.18)	1.56(1.09–2.24) **	1.41(0.94–2.13)	0.99(0.50–1.98)	0.73(0.32–1.65)
Upper tertile	0.65(0.46–0.91) *	1.33(0.84–2.11)	1.17(0.79–1.75)	1.06(0.66–1.73)	0.83(0.43–1.60)	0.58(0.25–1.35)
**Multidimensional Scale** **of Perceived** **Social Support**						
Low	1	1	1	1	1	1
Middle	0.35(0.26–0.46) ***	0.36(0.26–0.49) ***	0.24(0.17–0.35) ***	0.25(0.17–0.37) ***	0.44(0.28–0.67) ***	0.45(0.28–0.72) **
High	0.22(0.15–0.32) ***	0.22(0.14–0.34) ***	0.32(0.23–0.45) ***	0.30(0.21–0.43) ***	0.29(0.19–0.46) ***	0.30(0.18–0.49) ***

Note. CMD: common mental disorder. Odds ratio (OR); 95% confidence interval (CI); significant level * *p* < 0.05, ** *p* < 0.01, *** *p* < 0.001. ^a^ All the listed variables were mutually adjusted.

## Data Availability

Researchers who wish to access to the data of the cohort in the current study can contact S.-M.C. via email at siuming.chan@cityu.edu.hk with a methodologically sound proposal.

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
