# Peer review of "Inequalities in Psychiatric Morbidity in Hong Kong and Strategies for Mitigation"

_ijerph, 2022, doi:10.3390/ijerph19127095_

Round 1

Reviewer 2 Report

The paper is nice but the Authors did not put any relevant data about the covid 19 and psychiatric disorders. The paper also need a revision from a native English speaker.

I suggest to add and cite

Pavone, P; Ceccarelli, M; Marino, S; Caruso, D; Falsaperla, R; Berretta, M; Rullo, EV; Nunnari, G. SARS-CoV-2 related paediatric acute-onset neuropsychiatric syndrome. Lancet Child Adolesc Health ; 5(6): e19-e21, 2021 06.

Reviewer 3 Report

This study aims to explore social trends in psychiatric disorders using a large-scale epidemiological survey of mental illness. This study provides valuable data for proposing improvements to policies currently in place to combat mental illness.

Mental illness and mental health services are critical policies for national and local governments. I believe that the results of this study can be expected to be effectively utilized to improve these policies and make them more practical.

As for the statistical treatment, I believe that a clear rationale has been provided and that it has been fully considered.

Major Comments

The data analyzed in this study were from a large-scale epidemiological survey of mental illness conducted approximately ten years ago. The authors state in the Limitations of the Study part that the data used in this study may not reflect the most current regional conditions. While it is an inescapable fact that survey research is subject to time lags, it is preferable to use the most recent data or to add comparisons with the most recent data, if possible.

In the final part of the introduction, the authors also present the most recent data on mental illness. Still, they do not explicitly make direct comparisons with the data used in this study.

Given the use of data from approximately ten years ago, I believe it is essential to clarify changes over time in disease structure, etc., from the perspective of past and current data, as a complement to past data.

Reviewer 4 Report

This paper has an extensive work and it is an important Topic of investigation.  we congratulate the authors for the exhaustive work, however, a brief theoretical frame needs to be included to support the research and hypotheses should be stated.  

Round 2

Reviewer 1 Report

The revised manuscript answered my comments and concerns. It is now ready for publication.